# *Staphylococcus epidermidis* RP62A’s Metabolic Network: Validation and Intervention Strategies

**DOI:** 10.3390/metabo12090808

**Published:** 2022-08-28

**Authors:** Francisco Guil, Guillermo Sánchez-Cid, José M. García

**Affiliations:** Grupo de Arquitectura y Computación Paralela, Universidad de Murcia, 30080 Murcia, Spain

**Keywords:** *Staphylococcus epidermidis*, metabolic network validation, minimal cut sets, knock-outs, systems biology

## Abstract

Increasingly, systems biology is gaining relevance in basic and applied research. The combination of computational biology with wet laboratory methods produces synergy that results in an exponential increase in knowledge of biological systems. The study of microorganisms such as *Staphylococcus epidermidis* RP62A enables the researcher to understand better their metabolic networks, which allows the design of effective strategies to treat infections caused by this species or others. *S. epidermidis* is the second most commoncause of infection in patients with joint implants, so treating its proliferation seems vital for public health. There are different approaches to the analysis of metabolic networks. Flux balance analysis (FBA) is one of the most widespread streams of research. It allows the study of large metabolic networks, the study their structural properties, the optimization of metabolic flux, and the search for intervention strategies to modify the state of the metabolic network. This work presents the validation of the *Staphylococcus epidermidis* RP62A metabolic network model elaborated by Díaz Calvo et al. Then, we elaborate further on the network analysis’s essential reactions. The full set of essential reactions (including a previously unobserved one) was computed, and we classified them into equivalence classes. Some proposals to intervene in the network and design knock-outs by studying minimal cut sets of small length are also introduced. In particular, minimal cut sets related to the medium (including exchange reactions associated with medium metabolites) have been computed. In this sense, the unique external MCS (composed of cysteine and sulfate ion) has been found, and all hybrid MCS (based on knocking out both internal and exchange reactions) of length two have also been computed. The paper also points out the possible importance of these new intervention strategies.

## 1. Introduction

During the last two decades, the study of biological systems from a holistic point of view, together with the application of improvements in laboratory techniques and computational resources, has allowed a significant development of knowledge of living organisms at the molecular level. Cellular metabolism can be studied as a set of reactions that occur in the cell and allow its development and activity. These reactions are coordinated to maintain the balance of the cellular metabolism, also known as cellular homeostasis. With the assistance of high-performance computing, the metabolic network can be examined better to understand the overall activity of a whole cell. The metabolic network enables the generation of large amounts of omic data that allow identifying and quantifying biological molecules to discover the connections between genotype and phenotype. Then, researchers can make predictions of hypothetical states of the metabolic network by generating metabolic fluxes at different levels of the biological system.

Due to a large amounts of omic data which are continuously generated, the most successful approaches are those known as data-driven ones. These are based on combining omics data and computational methods to generate parts of or complete metabolisms called genome-scale metabolic models (GSMMs) [1]. GSMMs are mathematical representations that describe complete sets of stoichiometry-based mass-balanced metabolic reactions in an organism using gene–protein rule (GPR) association. GSMMs simulate how metabolism regulates and responds to changes, endogenous or exogenous, as mutations or changes in environmental conditions. The use of GSMMs has been successful in many applications, including the discovery of drug targets in pathogenic organisms, the prediction of enzyme functions, the analysis of multiple reactomes, and modeling cell-to-cell interactions in studying human diseases [2].

After the generation of GSMMs, the network has to be analyzed. To this end, the modeling approaches can be divided into two frameworks: constraint-based modeling (CBM) and kinetic modeling. Constraint-based modeling offers a representation of the metabolic network based on structural and stoichiometric aspects of the network with the aim of simulations of different states of metabolic flux and thus predicting phenotypes. On the other hand, the kinetic model aims to study the variations in metabolite concentrations over time through parameters related to the kinetics of chemical reactions and the concentration of enzymes in the metabolic network. Due to the nature of both frameworks, the first is used to study large-scale metabolic networks in a steady state; the second is restricted to networks of reduced size or parts of more extensive networks. Both frameworks are compatible and can be used to obtain a complete view of the model and draw more well-founded conclusions [3].

Focusing on CBM, one of the main tools to develop the analysis is flux balance analysis (FBA), a computational method to predict flux distributions while optimizing (maximizing or minimizing) a given cellular function or a combination of them [4]. Inside FBA there are plenty of tools, but in this study, we are mainly concerned with minimal cut sets (MCS). This tool tries to determine the minimal interventions necessary through minimum sets of deletions to be carried out in a metabolic network at the protein or gene level to generate phenotype knock-outs [5].

In this work, an analysis of the GSMM model of the *Staphylococcus epidermidis* RP62A, a pathogen, proposed by Teresa Díaz Calvo et al. [6], has been performed. For the first time, they presented the biological network model with fixed equations in their paper by observing the real behavior of the pathogen in the laboratory. Their paper attracted us due to the importance of *Staphylococcus epidermidis* as a pathogen. Therefore, we thought we could check and validate the proposed model and extract from the network model some intervention strategies to produce knock-outs of the pathogen.

*S. epidermidis* is a human opportunistic pathogen known to cause infections in medical implants (joints and heart). It is a facultative anaerobic bacterium with a Gram-positive coccus phenotype that produces a biofilm that, in the absence of disease, is located in collagen-rich regions such as the skin and mucous membranes. It belongs to the *Staphylococcus* genus, a vast group with well-known species such as *Staphylococcus aureus*, known for causing respiratory infections, among others. Concretely, prosthetic joint infections (PJI) are a recurring problem for arthroplasty and knee replacements. This type of infection occurs during the operation due to contamination of the material or the implant. PJI occurs when bacteria manage to adhere to the surface of the prosthetic implant and colonize it through the production of a biofilm. Most PJIs are of monomicrobial origin; the *Staphylococcus* genus was the first etiological agent discovered in PIJs (*S. aureus* and *S. epidermidis*, in order of incidence). Arthroplasty allows millions of patients relief from their pain and to recover or improve the mobility and functionality of the operated area. However, it is common for the diagnosis of PJI to be made when it is already chronic. At this point, surgical replacement of the implant is the only treatment with guarantees of success, although relapse is not unlikely [7]. The economic impact of PJIs is significant. Arthroplasty is a frequently performed procedure, and due to the cumulative effect of the aging population and increased life expectancy, it has a growing incidence. Therein lies the importance of studying the metabolic network of one of the main aetiological agents (*S. epidermidis*) to find solutions to prevent the growth of this bacterium (drugs, recombinant technologies, etc.). Finding feasible and safe solutions would improve patients’ quality of life and reduce the economic impact on the health system.

There are several ways to intervene in the metabolic network. The first way of doing so is by knocking out one (or several) internal reactions. In this approach, the minimal interventions are provided by the essential reactions (those active in any non-trivial state of the network). When canceling one of these reactions, the cell must die (internal cut sets). A second way is removing (or limiting) some of the available nutrients. This limitation can be simulated by knocking out exchange reactions associated with external metabolites (external cut sets). Finally, a third approach (a mix of the previously mentioned ones) is based on knocking out both internal and exchange reactions (hybrid cut sets).

The main goals we try to cover in our work are the following:Replicating the results obtained from Díaz Calvo et al. [6], validating the model, and discussing improvements to the calculations performed in the original work.Providing a better understanding of the *Staphylococcus epidermidis* RP62A metabolic network.Proposing new minimal interventions to the metabolic network to feasibly eliminate *S. epidermidis*.

## 2. Material and Methods

### 2.1. Constraint-Based Modeling

A metabolic model is defined by metabolites, the reactions that interconvert them (described by a fixed stoichiometry), and the fluxes that quantify the extent to which each reaction intervenes. The different components that make it up should be identified to establish a metabolic model.

Let R and M be the sets of reactions and metabolites of the system, and denote by *m* and *n* the numbers of metabolites and reactions in the network. Each reaction would convert specific amounts of some metabolites into amounts of other metabolites. This information is provided by the stoichiometric coefficients of the reactions that can be summarized in a matrix $S\in M_{m\times n}\left(R\right)$.

Each possible state of the network can be represented by a vector $v\in R^{n}$, where the component $r_{i}$ indicates the amount of flux through reaction $r_{i}$, and a vector $x\in R^{m}$ represents the concentration of each metabolite $m_{j}\in M$. The variation in concentrations of the metabolites is summarized in Equation (Equation 1)
(1)$$\frac{dx}{dt}=S\cdot v$$

The state of equilibrium in which the concentrations of internal metabolites of the metabolic network do not change over time (production and consumption of metabolites by biochemical reactions are equal) is called the steady-state constraint [8]
(2)S·v=0

Observe that, in this formulation, only internal metabolites must be included as rows in the stoichiometric matrix *S*.

Considering the thermodynamic directionality, the reactions of a metabolic network can be classified into irreversible and reversible. A reaction $r_{i}\in R$ is said to be irreversible if it can only carry flux in one possible direction. Irreversible reactions from a subset Irr⊂R. Reversible reactions are those that can carry flux in both directions.

The restrictions on the sign of the flux of the irreversible reactions are called the thermodynamic constraint (it will be supposed that all irreversible reactions always occur in the positive direction [9]):(3)$$r_{i}\geq 0\;\forall r_{i}\in R$$

A mode (or state) of the network is a flux vector satisfying Equations (Equation 2) and (Equation 3). Two states of the network are considered equivalent if one is a non-negative multiple of the other. With a slight abuse of notation, they are always identified as the same mode. The set of all possible network modes (the flux cone of the metabolic network) is denoted as *C*.
$$C=\{v\in R^{n}\;|\;S\cdot v=0,v_{i}\geq 0\;\forall r_{i}\in Irr\}$$

Given a mode v∈C, its support, supp(v), is defined as the set of reactions that appear with non-zero flux in *v*. That is,
$$supp\left(v\right)=\left\{r_{i}\in R\;|\;v_{i}\neq 0\right\}$$

A reaction *r* is said to be essential if r∈supp(v)∀0≠v∈C. That is, *r* is active in any non-trivial state of the metabolic network. Having the essential reactions implies knowing which reactions are necessary for the organism’s life. On the other hand, a reaction is said to be blocked if it is always inactive.

Given a set of target reactions, T⊂R, a cut set for *T* is a set of reactions whose inactivation induces the inactivation of all the reactions in *T*. Formally, if T⊂R is a set of target reactions, a subset S∈R is called a cut set for *T* if S∩T=Ø and ∀v∈C,supp(v)∩S=Ø⇒supp(v)∩T=Ø. A cut set *S* is a MCS if $∄\;S^{\prime }\subset S$ such that $S^{\prime }$ is also a cut set for *T* (see [10]).

Thus, an essential reaction is just an MCS of length 1 for the whole set of reactions.

Another important concept is that of the dependencies of reactions. Given two unblocked reactions $r_{i}$ and $r_{j}$, it can be said that:$r_{i}$ (partially) implies $r_{j}$ if whenever $r_{i}$ is active, $r_{j}$ must also be active.$r_{i}$ is equivalent to $r_{j}$ if $r_{i}\to r_{j}$ and $r_{j}\to r_{i}$.$r_{i}$ (totally) implies $r_{j}$ if $r_{i}\to r_{j}$ and there is a constant 0≠c such that in any state in which $r_{i}$ is active, $v_{j}=c\cdot v_{i}$ is satisfied.

**Flux balance analysis.** FBA performs the network analysis of a model. In this framework, adding a function transforms our steady-state and thermodynamic constraints into a linear programming (LP) optimization problem. Concretely, if $\left\{a_{i}\right\}\in R^{n}$ is any sequence of constants, the linear function on the fluxes $f\left(v\right)=\sum_{i=1}^{n}a_{i}\cdot v_{i}$ can be considered to pose the associated LP problem:(4)$$\begin{array}{ccccc} & \mathrm{Minimize}\; & & f(v)\\ & \mathrm{subject}\;\mathrm{to}\; & & S\cdot v=0\\ & \; & & v_{i}\geq 0 & \;\forall r_{i}\in Irr \end{array}$$

The obtained solution is a network mode. Using different additional constraints and functions, different scenarios in the metabolic network can be emulated (see [9]).

### 2.2. *Staphylococcus epidermidis* RP62A Model

The original model of *Staphylococcus epidermidis* RP62A was implemented by Díaz Calvo et al. [6] in the Systems Biology Markup Language (SBML (https://sbml.org/), a free and open data format for computational systems biology used by thousands of people worldwide. The authors used the ScrumPy metabolic modeling package, a widely used framework for metabolic network analysis.

The GSMM of *S. epidermidis* RP62A consists of 893 reactions, plus 97 transport reactions, 864 internal metabolites, and 74 external metabolites. The curated model has mass and conservation energy balanced. Biomass generation is represented by pseudo-transporters (exchange reactions) to simulate the export of biomass precursors. A simplification of the model is shown in Figure 1. There is no gene information in the SBML file provided by the creators.

Simplification of the model for *Staphylococcus epidermidis* RP62A. On the left is the modified HHW culture medium (Hussain, Hastings, and White) [11] which represents the uptake of nutrients incorporated by the metabolic network via pseudo-transporters (spotted arrow); in the center, the internal metabolic network is represented by a big square with the principal pathways on the left and the ATPase reaction on the right (represents ATP maintenance cost); on the right is the representation of the 48 biomass precursors declared as external metabolites exiting the metabolic network via pseudo-transporters. In this image, all biomass precursors have been gathered in a biomass lumped reaction.

**Biomass composition.** The model simulates biomass production (mmol gDCW^−1^) through 48 precursors that are declared as external metabolites by using transport pseudo-reactions. Biomass consists of biofilm-forming precursors and planktonic cell components, such as amino acid residues, deoxynucleotides, nucleotide triphosphate, cell wall/membrane components, and soluble metabolites. The fluxes of the reactions associated with these precursors were measured experimentally in continuous culture at a steady state. A complete explanation of the biomass composition can be found in the supplementary material of [6].

Denote by $R_{B}\subset R$ the subset formed by those transport pseudo-reactions. For each reaction, $r_{i_{j}}\in R_{B}$, a constant $b_{i_{j}}$ has been experimentally obtained so that any vector flux v∈C must fulfill
$$v_{i_{j}}=b_{i_{j}},\;\forall r_{i_{j}}\in R_{B}$$

**Culture medium.** A set of nutrients represents the medium made up of ions, amino acids, and secondary metabolites that are incorporated into the metabolic network through transport reactions and is measured in gDCW^−1^ units [12]. This is the HHW medium described by Hussain et al. [11] which has been modified by Diaz Calvo et al. [6] by adding asparagine and eliminating unnecessary components of the medium. This medium contains all amino acids except glutamine [6].

Denoting by $R_{M}$ the subset of these transport reactions and by $m_{k_{l}}$ the maximum value of the transport reaction $m_{k_{l}}$ (provided in the paper mentioned above), the following additional constraint for the model is obtained.
$$0\leq v_{k_{l}}\leq m_{k_{l}},\;\forall r_{k_{l}}\in R_{M}$$

**ATP maintenance cost and Specific Growth Rate.** Measures of growth ($Y_{ATP}$) and non-growth ATP ($m_{ATP}$) maintenance costs, growth-associated ATP maintenance cost (60 mmol gDCW^−1^), and non-growth-associated ATP maintenance cost (8 mmol gDCW^−1^h^−1^), respectively, were taken by experimental procedures.

In this model, an ATPase reaction, $r_{ATPAse}$, is introduced, and the flux through this reaction must fulfill
$$v_{ATPAse}=A=Y_{ATP}\cdot \mu +m_{ATP}$$

The value μ has been experimentally calculated with different medium compositions. This parameter is used to describe the dynamic behavior of microorganisms, and it is measured in gDCW^−1^h^−1^. The specific growth rate (μ) must be measured in cell growth kinetics experiments under continuous-medium conditions. When growth rates are obtained experimentally, they may be provided in h^−1^ units [12].

The parameter values were taken experimentally in wet laboratory conditions by Díaz Calvo and collaborators.

An LP program was formulated to obtain the optimal total flux as a minimization problem. Due to the presence of reversible reactions, to obtain this minimal flux state, the function $f\left(v\right)=\sum_{i=1}^{n}\left|v_{i}\right|$ is minimized while taking into account the additional constraints previously indicated.

Thus, the optimization problem can be stated as follows:(5)$$\begin{array}{cccc} & \mathrm{Minimize}\; & & f\left(v\right)=\sum_{i=1}^{n}\left|v_{i}\right|\\ & \mathrm{subject}\;\mathrm{to}\; & & S\cdot v=0\\ & \; & & 0\leq v_{i}\;,\;\forall r_{i}\in Irr\\ & \; & & v_{i_{j}}=b_{i_{j}}\;,\;\forall r_{i_{j}}\in R_{B}\\ & \; & & 0\leq v_{k_{l}}\leq m_{k_{l}}\;,\forall r_{k_{l}}\in R_{M}\\ & \; & & v_{ATPase}=A \end{array}$$

Observe that, in this problem, the function *f* to be minimized is not linear. However, if all the reactions are irreversible, then *f* can be rewritten as $f\left(v\right)=\sum_{i=1}^{n}v_{i}$. Again, and due to the first two constraints, each solution to this FBA problem is particularly a network mode.

### 2.3. Our Experimental Approach

In our approach, the analysis of the model was performed using the COBRApy (Constraint-Based Reconstruction and Analysis in Python) package [13] working within a Jupyter-Notebook [14] with Python 3.6 as the kernel [15]. For solving the associated LP problems, we have used Cplex version 12.10 [16].

All the Python computer programs have been run in the Gacop’s Cluster formed by several x86-64 computing nodes connected with an internal Gigabit Ethernet Network running Centos GNU/Linux 8.2. Cluster support has been provided by the Research Group of the High-Performance Computer Architecture (GACOP) of the University of Murcia (Spain). After finishing the importation of the model by COBRApy, an analysis of the structural properties was performed.

The provided *S. epidermidis* model was successfully imported in COBRApy after solving a few issues (a detailed explanation of the importing process is provided in Appendix A). It should be noted that the advice of the former research group was a great help when the first experiments with the model began. However, a few difficulties were found while dealing with the calculations performed during that work.

The GSMM of *S. epidermidis* RP62A consists of 893 reactions plus 97 transport reactions and 864 internal metabolites, and 74 external metabolites.

The model contains 74 metabolites that are declared as external. There also are another 277 metabolites (dead-end metabolites) that behave as external ones in the sense of having no reaction producing or consuming them. These metabolites can be considered "blocked", and their presence in any reaction implies that these reactions are blocked. The average percentage of blocked metabolites in other models has been computed by checking curated models similar in size taken from the BIGG database [17]. The average amount of blocked metabolites in these models is around 40–50%, which fits with the proportion of our case model (≈30%). An analysis of the number of blocked metabolites in this model compared with several other models from BIGGs is available in Appendix A.

Regarding the directionality of reactions, there are 405 reversible reactions and 585 irreversible ones. An unfolding of the reversible reactions of the model (substituting each reversible reaction with two irreversible ones that represent the two possible directions) was done to avoid the appearance of absolute values while calculating modes of minimal total flux. This unfolding step turned our optimization problem posed in Equation (Equation 5) into a linear one, so we could use standard LP methods. After unfolding the metabolic network, the new model had 864 metabolites and 759 reactions (all irreversible). The original and decoupled SBML models can be downloaded from https://github.com/biogacop/Sepidermidis_Analysis.

We have also explored two additional ways to solve the original optimization problem (see Appendix A for details):Mixed-integer linear programming (MILP). Many solutions are obtained depending on the parameter that controls the optimization procedure.Formulating a new optimization problem by creating a lumped biomass pseudo-reaction that gathers all its precursors in a simplified way. The modified model can also be downloaded from https://github.com/biogacop/Sepidermidis_Analysis.

Finally, we examined some structural properties of the model. We obtained the model’s percentage ratio of the dead-end metabolites (29.53%). This value can be considered quite reasonable compared to models of similar sizes from the BIGG database [17], which is known for having carefully crafted and curated models. This comparison can be found in Appendix A.

After fixing the minimal total flux as an additional constraint, we also examined the possible oscillations of the concentration values of the culture medium’s component under the minimal total flux condition. These variations were as expected: they ranged between 0 and a value corresponding to the maximum values for the corresponding exchange reactions. An exception was found for a few amino acids (isoleucine, leucine, lysine, methionine, tyrosine, and phenylalanine) and niacin (vitamin B3). In these cases there were significant differences between the minimum and maximum values that were incorporated into the metabolic network and those obtained in our analysis. Isoleucine, leucine, tyrosine, and phenylalanine were halved, lysine was doubled, methionine was increased by one order of magnitude, and niacin was increased by two orders of magnitude (Appendix A contains a table with all those oscillation values).

Figure 2 shows the pipeline used for the validation of the previous model [6]. The pipeline also describes the new experiments we have done:

## 3. Results

### 3.1. Validation of the Model

The original paper posed an optimization problem to obtain a minimum total flux state of the metabolic network using Equation (Equation 5) and replicating the conditions and restrictions given in the model. They obtained a unique solution consisting of 227 reactions (excluding transport pseudo-reactions), of which 127 were essential. The total flux value produced by this solution was not reported.

In our work, we have successfully imported the model using COBRApy. We have replicated the optimization problem with the same conditions and restrictions. As the optimal solution is not unique, we were not able to find the exact solution reported in the original paper. Instead, we calculated some optimal solutions. In this study, we worked with 108 optimal solutions, their support size ranging from 295 to 344 reactions (from 218 to 268 reactions if transport pseudo-reactions are discarded). We found 130 reactions (not counting the transport reactions) present in all the 108 obtained solutions and 553 different reactions that appeared in support of at least one solution.

Moreover, we used two other techniques to solve the optimization problem: MILP and a new optimization problem by creating a lumped biomass pseudo-reaction that gathers all its precursors. The solutions obtained in both methods are very similar and entirely consistent with those obtained with the previous LP method.

In their paper, Díaz Calvo and colleagues also calculated the impact of removing individual amino acids. The impact was obtained by Euclidean distance between the flux vectors obtained by solving the original FBA problem and the one obtained after removing each amino acid. We decided not to replicate this part of the research because, as previously stated, the solution associated with the optimal flux value is not unique. Thus, it makes no sense to estimate the impact of the elimination of an amino acid by just comparing two of those solutions.

Regarding the concentrations of the components of the culture medium, the flux of the reactions that incorporate them into the metabolic network has been analyzed. We looked for the optimal total flux conditions subject to the minimization objective function. We observed that the maximum values consumed for some components (isoleucine, leucine, lysine, methionine, tyrosine, phenylalanine, and niacin (vitamin B3)) in minimal flux condition are lesser that the maximum amounts available in the culture medium (a table describing this can be consulted in Appendix A). After all, the models represent living beings, and these do not always incorporate all the nutrients available in the environment, but only those that are required for the optimal functioning of the metabolism.

### 3.2. Minimal Intervention Strategies

#### 3.2.1. Internal Minimal Cut Sets

Regarding the essential reactions, we found 128 ones (excluding the ATPase reaction). This value is nearly identical to the one obtained in the original paper (127 essential reactions). The new essential reaction we found is the DIACYLGLYKIN−RXN one.

All possible implications among essential reactions have been calculated and used to obtain equivalences. It is worth noting that there appear trivial equivalences (those given by a coupling of reactions involving a metabolite, such as $r_{0}\to m\to r_{1}$) and non-trivial ones (the causal relationship is not known at first glance).

We have classified the essential reactions (internal cut sets of length 1) into 35 equivalence classes. Seventeen essential reactions have no equivalent ones, and the other 111 are located in 18 equivalence classes with more than one element.

After obtaining these equivalences, we also studied their implications to distinguish between direct and indirect implications. Let us start by observing that if $r_{i}$ is a cut set for a target set of reactions *T* and $r_{j}$ is a cut set for $r_{i}$, then $r_{j}$ is also a cut set for *T*. Thus, we can think of the set of essential reactions (cut sets of length 1 for the biomass precursors) as chains formed by cut sets of those precursors, cut sets for these (primary) cut sets, and so on. This distinction allows us to differentiate between reactions whose deletion directly blocks a biomass precursor (direct implication) and those whose deletion blocks a reaction that directly implies a biomass precursor (indirect implication). This classification leads to a better explanation of the essential reaction behavior.

Two subsets of essential reactions are significant here: the set of direct implications and final ones (that is, cut sets for the biomass precursors that have no other cut sets of length 1). Regarding the direct implications, we found 20 of them (the complete list of implications is provided in Appendix A). On the other hand, we saw the following 12 final essential reactions:Five evident implications: the ATPase reaction, the NAD cofactor activation reaction, the hydrolysis of pyrophosphate (which has many secondary implications), RXN66-532 (alpha-D-phosphohexomutase, catalyzes the interconversion between glucose-6-phosphate and alpha-glucose-1-phosphate), and the phosphorylation of diacylglycerol, which participates in the glycolytic pathway.Seven other implications: We have itemized this part to avoid long sequencesNICONUCADENYLYLTRAN-RXN (nicotinate-nucleotide adenylyltransferase, adenylation of nicotinate mononucleotide to nicotinic acid adenine dinucleotide).RXN-12002 (UMP/CMP kinase, phosphorylates UMP to UDP).SHIKIMATE-5-DEHYDROGENASE-RXN (shikimate dehydrogenase (NADP+), catalyzes the reversible NADPH linked reduction of 3-dehydroshikimate to shikimate and involved in the biosynthesis of aromatic amino acids).HYDROXYMETHYLGLUTARYL-COA-SYNTHASE-RXN (hydroxymethylglutaryl-CoA synthase, participates in ergosterol biosynthesis by condensing acetyl-CoA with acetoacetyl-CoA to yield hydroxymethylglutaryl-CoA).IPPISOM-RXN (isopentenyl-diphosphate delta-isomerase, catalyzes the isomerization of isopentenyl pyrophosphate to isopentenyl pyrophosphate taking part in the biosynthesis of isoprenoids).PRPPSYN-RXN (ribose-phosphate diphosphokinase, involved in the chorismate synthesis pathway, which is part of the synthesis of aromatic-type amino acids).Palmitate_synth (palmitate synthase, yielding palmitate a saturated fatty acid which is a component of the cell membrane).

A detailed study of the structure of the essential reactions can be found in the Appendix A.

After studying these equivalences, we could also analyze their implications. This study led to distinguishing between different types of essential reactions, giving interesting additional information about them.

Remember that essential reactions can be viewed as MCSs of length 1 for the biomass precursors. Thus, we have also explored other ways of achieving the intervention in the metabolic network for these biomass precursors by finding MCSs of greater length.

#### 3.2.2. External Minimal Cut Sets

Focusing only on reactions associated with the importation of nutrients, we have found a new MCS of length two (composed of cysteine and a sulfate ion). Due to the importance of these two reactions, a deeper study of their impacts on the biomass precursors has been carried out. The results are included in Appendix A.

#### 3.2.3. Hybrid Minimal Cut Sets

As there are no more MCSs that consist only of exchange reactions, the next natural step was to calculate cut sets in which culture medium components intervene along with some internal reactions (hybrid cut set). In this sense (using a method similar to the well-known Berge algorithm [18]), 48 hybrid MCSs of length two formed by a medium reaction and an internal one were calculated. There was also a hybrid cut set of length two involving *sulfate* (which was not in the medium) and *RIB5PISOM-RXN*.

Continuing with the idea of finding ways to intervene in the network and taking into account that cysteine and sulfate are essential for the functioning of the network, hybrid cut sets were calculated in which at least one of the two elements was present in the cut set. In this sense, an additional MCS of length three containing cysteine has also been found. The complete list of computed MCSs can be downloaded from https://github.com/biogacop/Sepidermidis_Analysis. These results are also included in Appendix A.

## 4. Discussion

### 4.1. Model Validation and Best Practices

It has been verified that the model created by Díaz Calvo et al. is interesting and has quite good quality for being so recent. The model is in its first iteration, and it would be very good to improve aspects such as the standardization of the *ids* (identifiers) and *names* of both metabolites and reactions.

Even though the solution to the optimization problem posed is not unique, it was able to obtain solutions that are very similar to that of Díaz Calvo et al. However, it was not possible to check if we obtained precisely the same solution as in the reference due to the lack of the total flux value. Even if the comparison between solutions cannot be completely accurate, we consider that the model has been validated by reproducing these results.

The minimization problem has also been studied from a more technical point of view. The same metabolic network flux minimization problem has been solved using MILP. We observed that the calculated solutions are similar to those obtained using LP methods. Once again, getting multiple solutions showed that the network has different states for the same conditions. Regarding the inclusion of a lumped biomass pseudo-reaction, the results obtained fall within the range of support sizes and the optimal obtained total flux.

We considered that the maximum limit of nutrients available in the medium that is the concentration of nutrients that *S. epidermidis* incorporates into its metabolism under steady-state conditions and continuous culture. Taking in lower concentrations of certain nutrients can be due to the conditions in which the state of the metabolic network is simulated. In short, the biological system is subjected to optimal minimum flux conditions (like a basal state). Therefore, it is logical to think that the body in basal conditions does not have the same energy and nutritional needs as when it is in ideal conditions. Even so, it is striking that these components of the medium are not fully incorporated, and yet the rest of the components are.

Finally, after our experience replicating the network model, we considered proposing a few points that should be taken into account when building a model (best practices):Homogenize the *id* and *name* naming system. Although, indeed, the notation of the model’s reactions, metabolites, and genes is usually automatized, when curating the model, a *naming* system must be taken into account to maintain the consistency of the model. Otherwise, it can lead the observer misinterpreting the model, making it difficult to interact with the metabolic network, leading to calculation mistakes.There are several possible reasons for a metabolite to be a *dead-end*: missing annotation, missing/absent exchange reactions, or simply that the reaction cannot carry flux at steady-state. In any case, this model is expected to reduce this ratio in future updates. Thus, it is important to declare the external and dead-end metabolites of the model explicitly. If there are differences between external metabolites that represent the limit of the metabolic network represented by the model and metabolites related to the representation of biomass and by-products, they should be mentioned.Detail on which databases (version) and organisms (assembly accession) the notation has been based on to build the model.Declare if there are pseudo-reactions or pseudo-metabolites and what functions they fulfill in the model.If a solution to an FBA optimization problem is given, explicitly declare the total flux obtained. The solution support should be included in the Appendix A if the solution is unique. This statement is for reproducibility purposes.

In conclusion, we have observed that the solution indicated in the original paper fits perfectly in the ranges of values we found. Therefore, we claim that the original model has been validated by our in silico experiments, and therefore, the model is correct.

### 4.2. Minimal Intervention Strategies

After the metabolic network model, some studies were performed to gain insights and properties of *Staphylococcus epidermidis’s* behavior.

Regarding the essential reactions, we found DIACYLGLYKIN−RXN, a previously unobserved essential reaction. We have found that this reaction is different from the other essential ones. All the other essential reactions are cut sets for at least a biomass precursor in the original model (leaving aside the additional restrictions given by the values of the biomass precursors). That is, for each essential reactions, $r_{i}$, there is at least a biomass precursor, $r_{j}$, such that for any mode v∈C, $v_{i}=0$ implies $v_{j}=0$. This implication is not true for r=DIACYLGLYKIN−RXN, since in this case, each biomass precursor can be active even if *r* is inactive. However, if *r* is inactive, it is not possible for all the biomass precursors to simultaneously achieve their corresponding values in the biomass composition. Let us remark that this reaction is catalyzed by the enzyme diacylglycerol kinase (ATP), a transferase that catalyzes the ATP-dependent phosphorylation of phospholipids for the cell membrane. Among the different phospholipids that this reaction can synthesize is diacylglycerol (DAG), one of the biomass precursors present in the *S. epidermidis* RP62A model. Therefore, it is a reaction that participates indirectly in synthesizing this precursor.

Additionally, we have studied some intervention methods in the metabolic network to find ways to design possible knock-outs for the network. In this sense, we have examined the set of essential reactions, including the previously unobserved one.

We have also explored other ways of achieving the intervention in the metabolic network for the biomass precursors. Focusing only on reactions associated with the importation of nutrients, we found a new MCS of length two (composed of cysteine and sulfate ion). Due to the importance of these two reactions, a deeper study of their impact on the biomass precursors was carried out. This MCS is interesting, as the *S. epidermidis* RP62A metabolic network is capable of synthesizing the amino acid cysteine, but requires sulfate. However, a sulfate MCS alone would not be viable, since the culture medium also provides exogenous cysteine to the microorganism so that it can be incorporated into the metabolic network through transporters. Consequently, even if the entry of sulfate into the metabolic network or even enzymes of the cysteine synthesis pathway were interrupted, the metabolic network would continue to carry the flux through another subset of reactions; in other words, the microorganism would not die. This MCS corroborates that there is no auxotrophy for cysteine. Figure 3 shows the sulfur pathway form the *S. epidermidis* RP62A model.

This MCS provides minimal intervention through the deprivation of cysteine and sulfate from the medium or by blocking the transporters that facilitate the entry of these components into the interior of the microorganism. This finding could lead to clinical research on treatments that try to reduce the proliferation of *S. epidermidis* in infections suffered by patients. Drugs such as antibiotics (experimented related to tetracycline water-soluble formulations) [19], and new antimicrobial biomaterials, such as nisin/polyanion layer-by-layer films [20], have been investigated, but none are related to the interruption of cysteine and sulfate at a medium level or metabolic pathways involving these two biomolecules.

Continuing with the study of the MCS of the network, we checked which reactions sulfate is essential for, and the same for cysteine. As expected, the reactions that specifically require cysteine are those that participate in pathways related to cysteine biosynthesis. The only one not associated with sulfate was the reaction involving ribose-5-phosphate isomerase, belonging to the pentose phosphate pathway.

The reactions of biomass precursors blocked in the absence of cysteine and sulfate were also explored. The results were as expected, since the blocked reactions were related to cysteine, methionine, and compounds necessary for synthesizing both amino acids and cofactors acetyl-CoA and CoA. The utilization of cysteine to synthesize methionine and vice versa is a process that occurs continuously through the transsulfuration pathway. Both are sulfur-containing amino acids and are of vital importance for synthesizing a wide variety of enzymes (e.g., cysteine can form disulfide bonds with other cysteine residues, which plays a crucial role in protein folding and protein structure). Another aspect of interest is the reactions in which cysteine and sulfate are involved.

An interesting aspect for future works would be to carry out the gene and functional annotation of the genes that code for the enzymes that carry out the reactions of interest.

## 5. Conclusions

During the last decade, there has been a revolution in systems biology and the study of increasingly sophisticated and realistic computational models. However, there is still a long way to go. Various models are emerging at different levels of study of biological systems, but these must be subjected to a rigorous curation process to reduce design flaws. However, despite efforts to standardize the format in which models should be written, there is much divergence when it comes to naming reactions and metabolites. In addition, there is also a lot of ambiguity in the terminology used to define key concepts when working with computational biological models. Hence, we suggest following the best practices provided in the discussion.

In this work, an exhaustive analysis of a model for *Staphylococcus epidermidis* was performed. Although it is not the only model that exists for *S. epidermidis*, it was one of the first. What makes this model interesting is that it was created and curated while taking into account empirical observations from experiments performed in the wet laboratory. Now, it has also been externally validated in silico by our research group. This provides great support to the model and offers guarantees of its quality. We have also been able to reproduce the first part of the results obtained by the creators of the model.

After the validation process, the set of essential reactions was computed. We also detected a previously unobserved essential reaction that is associated with the distribution of biomass precursors as described in the original model. This new essential reaction, DIACYLGLYKIN-RXN, is a reaction that participates indirectly in synthesizing diacylglycerol (DAG), one of the biomass precursors.

The strategy proposed in the original work was only based on the intervention of essential reactions. Although this strategy is a good option, we propose different minimal intervention strategies in which mixed interventions are used that involve the energy sources (nutrients from the culture medium) of the microorganism and non-essential reactions belonging to the metabolic network. In this sense, the unique external MCS (composed of cysteine and a sulfate ion) has been found. Finally, all hybrid intervention strategies (based on knocking out both internal and exchange reactions) of length two were also computed.

Further experimental investigations are now needed in clinically relevant conditions to study the possible applications of these intervention strategies.

## Figures and Tables

**Figure 1 metabolites-12-00808-f001:**
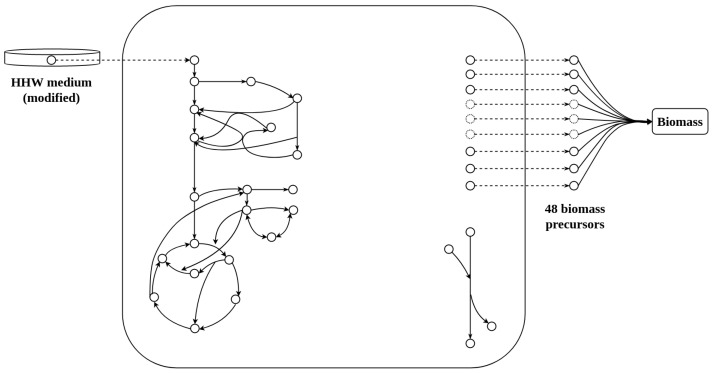
Model scheme.

**Figure 2 metabolites-12-00808-f002:**
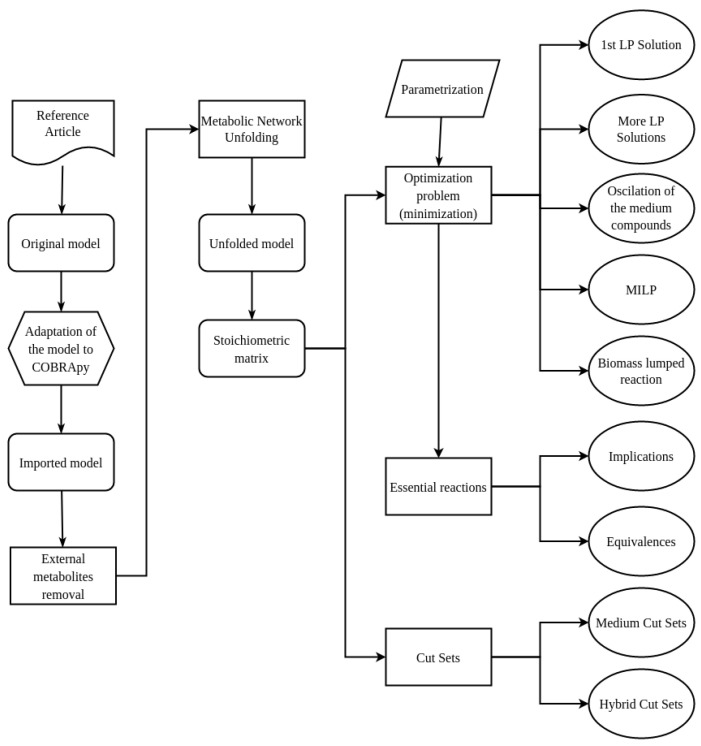
Pipeline for the analysis of the *S. epidermidis* RP62A model.

**Figure 3 metabolites-12-00808-f003:**
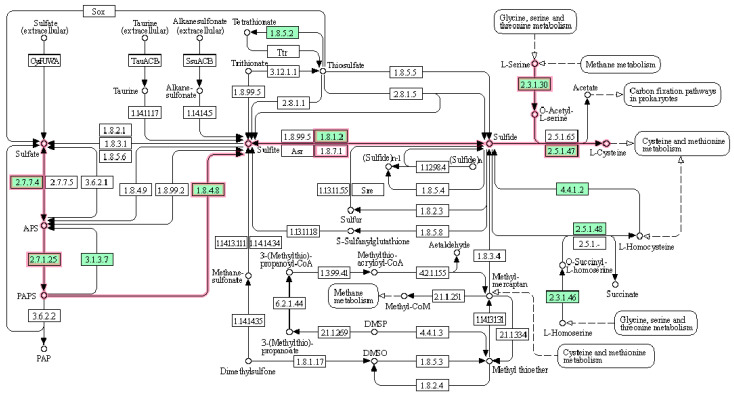
Sulfur pathway from *S. epidermidis* RP62A metabolism (KEGG database). The pathway for cysteine synthesis is highlighted in pink.

## Data Availability

Software is freely available at https://github.com/biogacop/Sepidermidis_Analysis.

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
