# Peer review of "Staphylococcus epidermidis RP62A’s Metabolic Network: Validation and Intervention Strategies"

_metabolites, 2022, doi:10.3390/metabo12090808_

Round 1
Reviewer 1 Report
The manuscript entitled “The Staphylococcus epidermidis RP62A metabolic network: Validation and intervention strategies” by Francisco Guil, Guillermo Sánchez-Cid, and José Manuel García is a well-written manuscript. However, please consider the following comments:
- The word “The” should be removed from the title of the manuscript and the title should be modified.
- The abstract should be rewritten with more details.
- Some sentences are too long and should be divided into two sentences.
- Some abbreviated words are mentioned in the text with their abbreviations two or three times like on page 2, in the first paragraph, the word Constraint-based modeling (CBM) is mentioned with its abbreviation two times. Once the word is abbreviated, it should be written in the abbreviated form after that.
- Please clarify the meaning of the essential reactions in the manuscript.
- The letter 2 should be written as two.
- The letters 5 and 7 should be written as five and seven.
- Where is the conclusion section?
Author Response
Dear reviewer 1,
We are glad to inform you that we have performed a major revision to the paper addressing your comments. We want to thank you for your insightful feedback and the exciting points raised, which helped us improve the paper, better understand our proposal, and make it brighter.
In the attached file we provide answers to the detailed issues, explaining how we have addressed them in the new version of the paper (highlighting these with comments when appropriate).

Reviewer 2 Report
In this manuscript, the validation of the Staphylococcus epidermidis RP62A metabolic network model elaborated by Díaz-Calvo et al., was made.
The following comments are made:
1. “S.epidermidis is a human 2. Separate S. from epidermidis
2. “gram-positive coccus”. uppercase Gram
3. Could you put the Methodology Section before Results?
4. “The results are included in the Supplementary Material”. You only mentioned that the results are in the Supplementary Material, but you never say in the text that part of the material is the one that should be reviewed. This appears throughout the text. Indicate what should be revised throughout the text.
5. The Results Section is not clear since you do not clearly mention what results you obtained.
6. The Discussion looks like the Results section. Put the results obtained in the Results Section and in the Discussion Section, discuss the results obtained.
7. Figure 1 is not cited in the text.
8. In the Discussion put how you could check yourn earlie models.
9. What is the contribution to the area of knowledge, apart from validating an earlier model?
10. What are your conclusions? There is no Conclusions section.
11. Check the references, it is necessary to put in the reference: “Place: San Francisco Publisher: Public Library Science”, as in 5. The same thing happens in other references, but not in all. Homogenize and review the way of putting the references.
Author Response
Dear reviewer 2,
We are glad to inform you that we have performed a major revision to the paper addressing your comments. We want to thank you for your insightful feedback and the exciting points raised, which helped us improve the paper, better understand our proposal, and make it brighter.
In the attached file we provide answers to the detailed issues, explaining how we have addressed them in the new version of the paper (highlighting these with comments when appropriate).

Round 2
Reviewer 2 Report
The authors heeded all suggestions.